# Analysis of functional brain connectivity in patient with end-stage kidney disease undergoing peritoneal dialysis using functional near infrared spectroscopy

**Chang Min Heo**[ID][1◉], **Jiyae Yi**[1◉], **Dong Ah Lee**[2], **Kang Min Park**[2], **Yoo Jin Lee**[1], **Sihyung Park**[1], **Yang Wook Kim**[1], **Junghae Ko**[1], **Aryan Yohanes Djojo**[ID][3], **Bong Soo Park**[ID][1]*

1 Department of Internal Medicine, Inje University College of Medicine, Busan, Korea, 2 Neurology Haeundae Paik Hospital, Inje University College of Medicine, Busan, Korea, 3 Department of Internal Medicine, Fatmawati Central General Hospital, South Jakarta, Indonesia

◉ These authors contributed equally to this work.
* H00245@paik.ac.kr

## Abstract

### Introduction

Neurological complications are common in patients with end-stage kidney disease (ESKD). However, the mechanisms underlying neurological complications of ESKD are poorly understood. Research on brain connectivity in patients undergoing peritoneal dialysis (PD) is limited. In this study, we aimed to examine alterations in functional brain connectivity in patients with ESKD undergoing PD compared to the control group using functional near infrared spectroscopy (fNIRS).

### Methods

We prospectively enrolled 20 patients with ESKD who had been receiving PD for more than 6 months and had no prior history of psychiatric or neurological diseases, along with 20 healthy controls. The fNIRS data were obtained using an NIRSIT Lite instrument. After processing all the data, we used Pearson correlation analysis to create a weighted connectivity matrix. Functional connectivity measures were derived from the connectivity matrix using graph theory. Functional connectivity measures were compared between the controls and patients with ESKD undergoing PD.

### Results

The average degree, average strength, and small-worldness were significantly lower in patients with ESKD undergoing PD than in the controls (9.333 [8.000～11.433] vs. 12.733 [9.600～13.400], p＝0.030; 6.865 [4.768～7.560] vs. 8.432 [6.593～9.432], p＝0.036; 0.836 [0.724～0.900] vs. 0.949 [0.882～0.972], p＝0.025, respectively).

**Data availability statement:** All relevant data are within the manuscript and its Supporting Information files.

**Funding:** This work was supported by the National Research Foundation of Korea (NRF) grant funded by the Korea government (MSIT) (No. RS-2023-00209722 to DAL) and the 2023 Inje University (research grant to CMH). The funders had no role in study design, data collection and analysis, decision to publish, or preparation of the manuscript.

**Competing interests:** The authors have declared that no competing interests exist.

## Conclusion

This study demonstrated significant alterations in functional brain connectivity in patients with ESKD undergoing PD, suggesting that functional brain connectivity is significantly reduced in patients with ESKD undergoing PD when compared with that in healthy controls.

## 1. Introduction

End-stage kidney disease (ESKD) is the final stage among the five phases of chronic kidney disease (CKD) and is characterized by a glomerular filtration rate (GFR) < 15 mL/min per 1.73 m$^2$. [1] Complications of ESKD include fluid retention, electrolyte abnormalities, cardiovascular disease, neurological abnormalities, chronic kidney disease-mineral bone disorder (CKD-MBD), and anemia. Neurological complications, such as cognitive deterioration, frequently occur in patients with ESKD. [2] The uremic toxins, anemia, vascular calcification, chronic inflammation, and intradialytic hypotension in patients with ESKD have been acknowledged as contributing factors for the development of neurological problems. [3]

Kidney replacement therapy, including hemodialysis (HD), peritoneal dialysis (PD), and kidney transplantation (KT), is the treatment of choice for patients with ESKD. [4] Because opportunities for KT are limited, most patients with ESKD undergo HD or PD. Compared to HD, PD results in fewer hemodynamic changes and better preservation of residual renal function. [5] However, the disadvantages of using glucose-containing dialysates include weight gain and hyperglycemia. As each dialysis method has its own characteristics, its impact on neurological complications varies.

Functional brain connectivity refers to the patterns of interactions and communication between different brain regions, and aids in the comprehension of how various brain regions interact to perform specific functions. Several modalities, including resting-state functional magnetic resonance imaging (rs-fMRI), [6] magnetic encephalography, [7] electroencephalography (EEG), [8] and positron emission tomography [9] can be used to investigate functional brain connectivity. Functional near-infrared spectroscopy (fNIRS) is a method for measuring functional brain connectivity and can precisely measure brain oxygen saturation. Recently, fNIRS has been used as an indicator of cerebral hemodynamics and functional brain connectivity. [10] fNIRS is a safe and non-invasive technique that utilizes near-infrared rays and has excellent mobility, superior temporal resolution compared to MRI, and superior spatial resolution compared to EEG [11,12].

In a recent study using rs-fMRI to understand the mechanisms underlying neurological complications in patients with ESKD, patients with ESKD showed aberrant functional brain connectivity compared to healthy controls. [13] One study compared structural and functional brain connectivity between PD and HD patients with ESKD using rs-fMRI and diffusion tensor imaging (DTI) and found that depending on the dialysis method, the brain connectivity changed differently compared to the healthy control group. [14] The different characteristics of PD and HD are thought to have

influenced the different patterns of brain connectivity. Furthermore, when comparing patients with ESKD with and without cognitive impairment using fNIRS, we observed that functional brain connectivity in the prefrontal brain network was compromised. [15] However, most studies have focused on patients with ESKD undergoing HD, and there is still limited research on functional brain connectivity in patients with ESKD undergoing PD.

Consequently, the purpose of this study was to examine, using fNIRS, the alteration of functional brain connectivity in patients with ESKD undergoing PD compared to a control group. We aimed to elucidate the mechanisms underlying the neurological complications in patients with ESKD undergoing PD.

## 2. Materials and methods

### 2.1. Participants

We enrolled 20 patients with ESKD undergoing PD between 5th October 2023 and 28th February 2024 at Inje University Haeundae Paik Hospital based on the following criteria: 1) clinically diagnosed ESKD, with a GFR < 15 ml/min/1.73 m$^2$, requiring renal replacement therapy; 2) received peritoneal dialysis for > 6 months; 3) absence of any prior neurological or psychiatric diseases; and 4) at least 18 years. In addition, 20 healthy participants aged > 18 years without a history of CKD or neurological or psychiatric disorders comprised the control group.

This prospective study was approved by the Institutional Review Board (IRB) of Haeundae Paik Hospital and all methods were conducted in compliance with the relevant guidelines and regulations. (IRB number: HPIRB 2023–04–004–006). Before the study began, all patients were informed of the research procedure and provided written informed consent. All patients underwent laboratory tests and the Korean version of the Montreal Cognitive Assessment (MoCA-K) to evaluate their cognitive function.

### 2.2. fNIRS data acquisition

The process of obtaining functional brain connectivity in patients with ESKD undergoing PD is shown in Fig 1. fNIRS data were obtained using an NIRSIT Lite device (OBELAB Inc., Seoul, Korea). [16] A portable, wireless, and wearable fNIRS

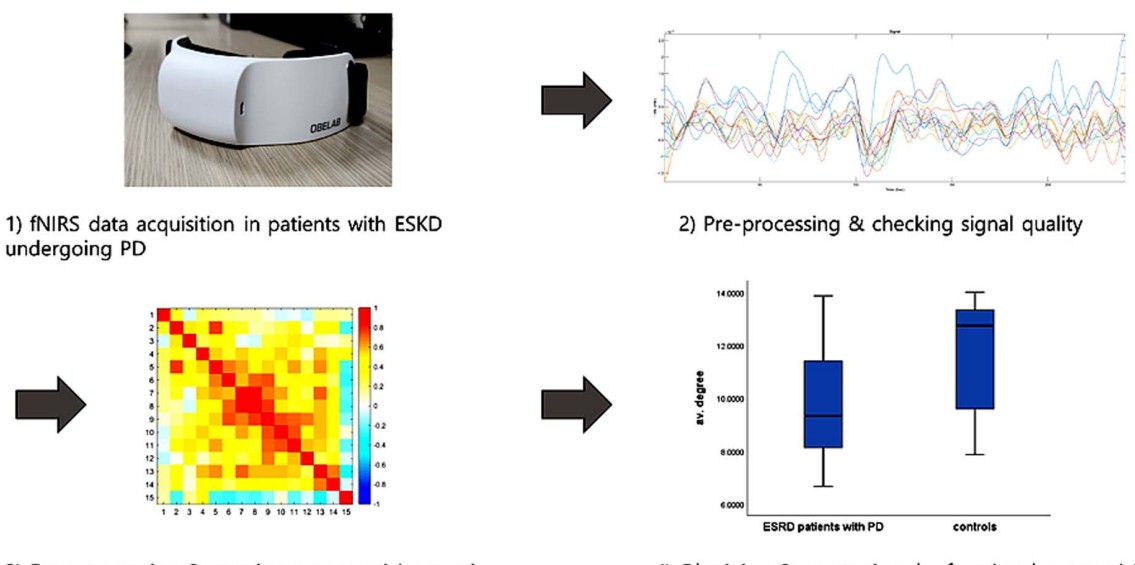

1) fNIRS data acquisition in patients with ESKD undergoing PD

2) Pre-processing & checking signal quality

3) Data processing & creating a connectivity matrix

4) Obtaining & comparing the functional connectivity measures

**Fig 1. Process of obtaining functional brain connectivity in patients with ESKD undergoing PD using functional near-infrared spectroscopy.**

device, NIRSIT Lite, determines the perfusion status in real time. This assay quantifies oxyhemoglobin (HbO$_2$) and deoxy-hemoglobin (HbR) by utilizing the disparity in their respective near-infrared light absorption rates. Thirteen detectors and five sources comprised the NIRSIT Lite. Fifteen channels were used to identify fNIRS signals in the prefrontal cortex. [17] The system uses wavelengths of 780 and 850 nm for near-infrared rays. The sampling rate used to determine the signals was 8.138 Hz. The acquisition of fNIRS data through the NIRSIT Lite was conducted during regular outpatient visits. To create the same environment for each patient, measurements were taken while the patients were in a resting state for 300 s while gazing at a tablet PC screen with a white cross on a black background.

### 2.3. Data processing and obtaining functional connectivity measures

The NIRSIT Lite Analysis Tool (version 3.2.4) was used for data analysis and connectivity matrix generation. The signal quality was assessed and the signal-to-noise ratio (SNR) anomalies were adjusted to 2.58, representing the criterion for channel rejection. A band-pass filter was implemented, featuring a low-pass threshold of 0.1 frequency and a high-pass threshold of 0.005 frequency. Pearson correlation analysis was used to process, export, and produce a weighted connectivity matrix for each patient.

Functional connectivity measurements were retrieved from the connectivity matrix via graph-theoretical analysis using BRAPH and MATLAB. [18] We calculated functional connectivity measures, including the assortative coefficient, average degree, average strength, characteristic path length, mean clustering coefficient, eccentricity, global efficiency, local efficiency, modularity, and small-worldness, in patients with ESKD undergoing PD [19,20].

### 2.4. Statistical analysis

We compared the demographic data and functional connectivity measures between patients with ESKD undergoing PD and controls using the chi-square test for categorical variables and the Mann–Whitney test for continuous variables. We applied the Benjamini–Hochberg correction to adjust the significance level in the statistical results of comparing functional connectivity measures. Spearman's correlation analysis was conducted to examine the relationships between clinical factors and functional connectivity measures. MedCalc® Statistical Software version 22.009 was used for all statistical analyses. (MedCalc Software Ltd., Ostend, Belgium; https://www.medcalc.org; 2023). Statistical significance was set at $p < 0.05$.

## 3. Results

### 3.1. Patient demographic and clinical characteristics

Twenty patients with ESKD undergoing PD and 20 healthy controls participated in this study. Table 1 shows the clinical characteristics of the patients with ESKD undergoing PD. The age and sex were not significantly different between patients with ESKD undergoing PD and healthy controls (60.5 [50.0~66.7] vs. 60.0 [50.7~62.2] years, $p = 0.495$; 11/20 (55%) vs. 13/20 (65%), $p = 0.519$; respectively). In patients with ESKD undergoing PD, the dialysis duration was 30.5 [14.2~42.2] months, Kt/V was 2.1 [1.7~3.4], systolic blood pressure (SBP) was 135.0 [118.0~146.0] mmHg, and diastolic blood pressure (DBP) was 73.0 [66.0~80.5] mmHg.

### 3.2. Comparison of the functional connectivity measures between patients with ESKD with PD and controls

Table 2 shows a comparison of the functional connectivity measures between patients with ESKD undergoing PD and controls. The results revealed significant differences in the average degree, average strength, and small-worldness between patients with ESKD undergoing PD and healthy controls (9.333 [8.000~11.433] vs. 12.733 [9.600~13.400], $p = 0.030$; 6.865 [4.768~7.560] vs. 8.432 [6.593~9.432], $p = 0.036$; 0.836 [0.724~0.900] vs. 0.949 [0.882~0.972], $p = 0.025$, respectively). The average degree, average strength, and small-worldness were lower in patients with ESKD with PD than in the controls.

### 3.3. Association between clinical factors and functional brain connectivity

Several clinical factors were significantly correlated with functional brain connectivity measures, including Kt/V and eccentricity (r = −0.591, p = 0.026), high density lipoprotein cholesterol (HDL-C), assortativity (r = −0.589, p = 0.021), hemoglobin and modularity (r = −0.517, p = 0.049), phosphate and eccentricity (r = 0.669, p = 0.009), and phosphate and small-worldness (r = −0.535, p = 0.049) (S1 Table, Fig 2). Other clinical factors, such as age, MoCA-K, SBP, DBP, years of education, dialysis vintage, body mass index, serum albumin, total cholesterol, triglyceride, low-density lipoprotein-cholesterol (LDL-C), hemoglobin, iron, ferritin, total iron-binding capacity, transferrin saturation, parathyroid hormone, and calcium showed no association with functional brain connectivity measures.

## 4. Discussion

In this study, we found a statistically significant difference in functional brain connectivity between patients with ESKD undergoing PD and healthy controls using fNIRS. Functional connectivity measures, namely average degree, average

**Table 1. Patients' demographic and clinical characteristics.**

| Variables | Patients with ESKD undergoing PD (n = 20) | Healthy controls (n = 20) | *p-value* |
|---|---|---|---|
| **Demographic data** | | | |
| Age, years | 60.5 [50.0~66.7] | 60.0 [50.7~62.2] | 0.495 |
| Sex, male | 11 (55) | 13 (65) | 0.519 |
| Dialysis duration, months | 30.5 [14.2~42.2] | | |
| Years of education, years | 12.0 [9.0~12.0] | | |
| MoCA-K | 24.0 [19.0~27.0] | | |
| Body mass index | 24.2 [21.5~26.8] | | |
| Kt/V | 2.1 [1.7~3.4] | | |
| Systolic blood pressure, mmHg | 135.0 [118.0~146.0] | | |
| Diastolic blood pressure, mmHg | 73.0 [66.0~80.5] | | |
| **Comorbidities** | | | |
| Hypertension | 18 (90) | | |
| Diabetes mellitus | 12 (60) | | |
| **Laboratory data** | | | |
| Hemoglobin, g/dl | 10.7 [9.4~11.6] | | |
| Iron, µg/dl | 98.5 [70.5~114.7] | | |
| Ferritin, ng/ml | 338.5 [117.7~471.2] | | |
| Total iron binding capacity, µg/dl | 257.5 [226.0~293.2] | | |
| Transferrin saturation, % | 35.6 [29.2~46.3] | | |
| Albumin, g/dl | 3.6 [3.2~3.9] | | |
| Total cholesterol, mg/dl | 151.5 [112.5~176.7] | | |
| Triglyceride, mg/dl | 122.0 [93.5~195.0] | | |
| High density lipoprotein-cholesterol, mg/dl | 41.5 [34.2~58.0] | | |
| Low density lipoprotein-cholesterol, mg/dl | 76.0 [48.0~99.5] | | |
| Calcium, mg/dl | 8.8 [8.2~9.8] | | |
| Phosphate, mg/dl | 5.2 [3.8~6.6] | | |
| Parathyroid hormone, pg/ml | 164.0 [108.0~291.5] | | |

Data are presented as the median with interquartile range or number with percentage.

PD: Peritoneal dialysis, MoCA-K: Korean version of Montreal Cognitive Assessment, Kt/V: Dialyzer clearance × time/distribution volume of urea

**Table 2. Comparison of the functional connectivity measures between patients with ESKD with PD and controls.**

| Network measures | Patients with ESKD undergoing PD (n=20) | Controls (n=20) | *p-value* | *Adjusted p-value* |
|---|---|---|---|---|
| Average degree | 9.333 [8.000~11.433] | 12.733 [9.600~13.400] | 0.003 | *0.030 |
| Average strength | 6.865 [4.768~7.560] | 8.432 [6.593~9.432] | 0.011 | *0.036 |
| Characteristic path length | 2.459 [1.579~4.180] | 1.939 [1.667~2.586] | 0.165 | 0.183 |
| Mean clustering coefficient | 0.587 [0.532~0.654] | 0.622 [0.555~0.708] | 0.433 | 0.433 |
| Eccentricity | 5.312 [4.142~8.223] | 4.026 [2.829~4.808] | 0.026 | 0.065 |
| Global efficiency | 0.548 [0.418~0.601] | 0.644 [0.542~0.707] | 0.033 | 0.066 |
| Local efficiency | 1.530 [1.231~1.765] | 1.746 [1.434~2.101] | 0.126 | 0.157 |
| Modularity | 0.182 [0.094~0.280] | 0.086 [0.027~0.181] | 0.037 | 0.061 |
| Assortative coefficient | 0.254 [−0.021~0.466] | −0.071 [−0.126~0.297] | 0.058 | 0.082 |
| Small-worldness | 0.836 [0.724~0.900] | 0.949 [0.882~0.972] | 0.005 | *0.025 |

Data are presented as the median value with interquartile range.

ESKD: End-stage kidney disease, PD: Peritoneal dialysis.

*Statistical significance ($p < 0.05$).

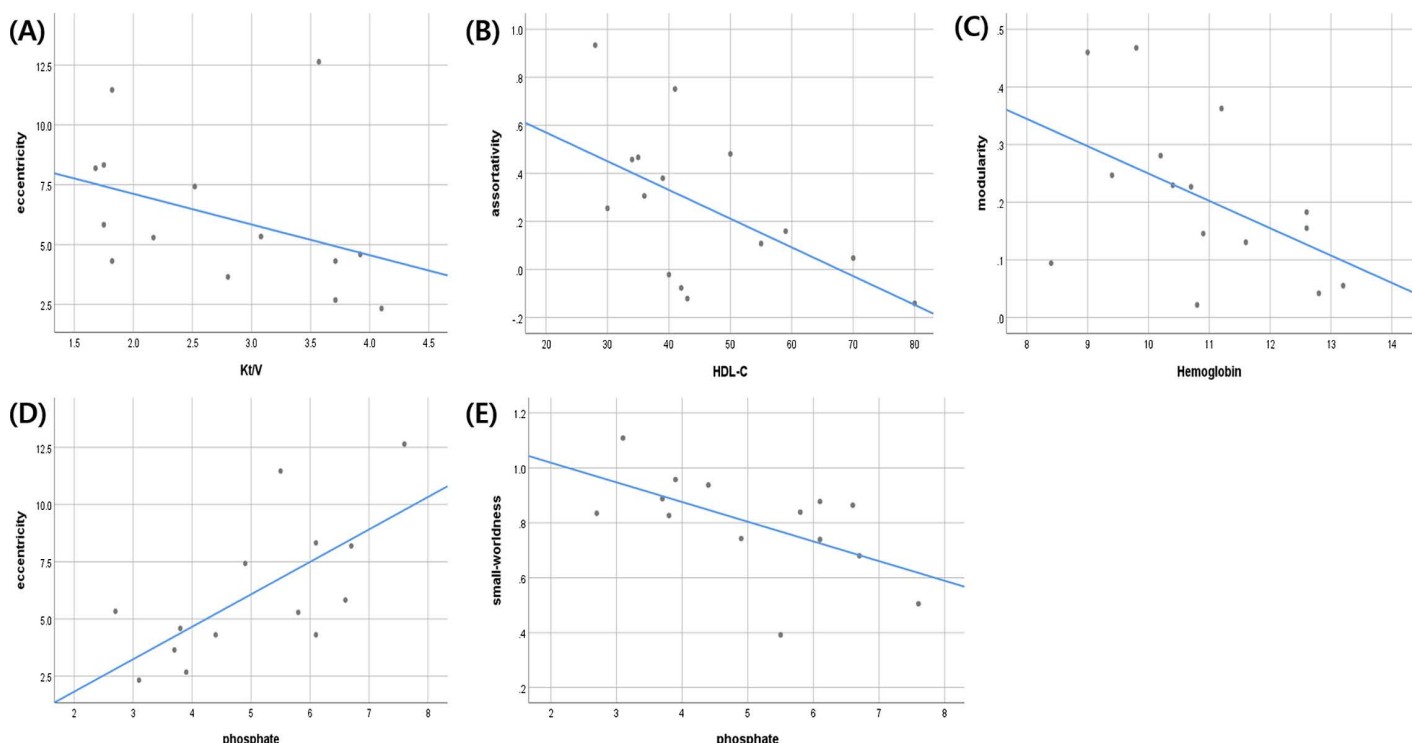

**Fig 2. Significant correlation between clinical factors and functional brain connectivity measures.** The results revealed a significant correlation between several clinical factors and functional brain connectivity measures, including Kt/V and eccentricity (r=−0.591, p=0.026) (A), HDL-C and assortativity (r=−0.589, p=0.021) (B), hemoglobin and modularity (r=−0.517, p=0.049) (C), phosphate and eccentricity (r=0.669, p=0.009) (D), phosphate and small-worldness (r=−0.535, p=0.049) (E).

strength, and small-worldness, were lower in patients with ESKD undergoing PD than in healthy controls. Furthermore, substantial correlations were observed between the clinical variables and measures of functional brain connectivity.

The prevalence of cognitive impairment, a neurological complication in patients with ESKD, varies in the literature, with reported rates ranging from 16% to 38%. [2] The exact mechanisms underlying the cognitive impairment in patients with ESKD are poorly understood. However, as previously mentioned, factors such as uremic toxins, anemia, vascular calcification, chronic inflammation, and intradialytic hypotension influence the development of cognitive impairment in patients with ESKD. [3] Several studies have indicated that cognitive functional impairment, a complication of ESKD, improves after dialysis in patients with ESKD. [21,22] In a study comparing cognitive function at baseline in patients with ESKD with cognitive function 1 year after the initiation of dialysis, it was shown that patients undergoing PD exhibit greater improvement in cognitive function compared to those undergoing HD. [23] HD is known to prescribe a more aggressive dialysis dose than PD and is associated with a higher frequency of hypotension during dialysis sessions. [24] However, PD is considered to be more continuous and effective in removing uremic toxins and managing anemia. [21] As it is thought that differences in dialysis modalities are thought to contribute to variations in cognitive function, investigating functional brain connectivity in patients with ESKD undergoing PD can be valuable for understanding the mechanisms underlying cognitive impairment.

fNIRS can precisely measure the brain oxygen saturation using near-infrared spectroscopy. When more blood flows to an area of the brain, the concentration of oxygenated hemoglobin increases, while that of deoxygenated hemoglobin decreases. fNIRS can measure variations in the levels of oxygenated and deoxygenated hemoglobin to determine changes in neural activity, because there is a strong association between changes in blood oxygenation and changes in neuronal activity. [25,26] Graph theory applied to the human brain is essentially a mathematical depiction of the actual brain architecture, where nodes represent different regions of the brain and edges represent the functional connections that connect them. [27] The application of graph theoretical analysis allows for a quantitative assessment of the effectiveness of brain networks. [19,20] Using graph theory and fNIRS, we identified alterations in functional brain connectivity in patients with ESKD undergoing PD compared to healthy controls.

Our results revealed significant differences in the average degree, average strength, and small-worldness between patients with ESKD undergoing PD and healthy controls. Degree and strength are the basic measures of connectivity. The average strength of a node in a network is its average weighted degree, while the average degree is a measure of the number of edges relative to the number of nodes. [27] A higher average strength may indicate stronger functional coupling between brain regions, suggesting increased coordination or communication among these regions. Therefore, decreased average degree and strength are accepted that functional connectivity between brain regions is diminished in patients with ESKD undergoing PD. A small-world network refers to a network topology characterized by short connections among a small number of nodes while also featuring relatively longer connections between certain nodes. [28] This structure enables efficient information transfer through local clustering and global connectivity. In the context of brain networks, small-worldness is associated with network efficiency and is contingent on the average shortest path length and global transitivity of the network. [14] Therefore, the decreased small-worldness in the functional brain connectivity of patients with ESKD undergoing PD indicates that the efficiency of the brain network has declined. In conclusion, due to the decrease in basic network measures and small-worldness, functional brain connectivity was significantly reduced in patients with ESKD undergoing PD compared to controls.

In this study, significant correlations were observed between clinical variables and measures of functional brain connectivity. Moreover, Kt/V and eccentricity demonstrated a negative correlation. Kt/V can be calculated as the dialyzer clearance multiplied by time divided by the distribution volume of urea, which represents the adequacy of dialysis. Eccentricity is defined as the maximum distance of one node from other nodes, and is negatively correlated with network integration. [29] Higher eccentricity in patients with ESKD undergoing PD indicates reduced functional integration. These results indicate that an increase in Kt/V is associated with an improvement in functional brain connectivity.

HDL-C levels were negatively correlated with assortativity. High HDL-C levels protect against cardiovascular diseases and play a role in preventing damage to the central nervous system. [30] Assortativity serves as a significant metric for assessing the resilience of a network by revealing whether nodes are densely connected to other nodes to comparable degrees or to those with significantly divergent degrees. [31] A positive assortativity value indicates an assortative network, whereas a negative value indicates a disassortative network. Compared to disassortative networks, assortative networks are more vulnerable to random attacks, and the transmission of information is considerably less efficient. [32] Thus, a decrease in HDL-C levels is associated with a assortative network, indicating that the functional brain network is inefficient.

Hemoglobin levels were negatively correlated with modularity. Anemia is a well-known risk factor for neurological complications in patients with ESKD. However, an association between high hemoglobin concentrations and cognitive impairment has been reported. [33] The result demonstrated a non-linear relationship between hemoglobin levels and cognitive function. The density of connections inside a module or community is measured by the term "modularity," which is a measure of a graph's structure. It measures functional segregation, which is the capacity of specialized processing to occur within clusters of densely interconnected brain regions. [27] In this study, we showed a negative correlation between hemoglobin levels and the functional segregation of the brain.

Phosphate levels were positively correlated with eccentricity and negatively correlated with small-worldness. In patients with ESKD, chronic kidney disease-mineral bone disorder (CKD-MBD) occurs as a complication, and hyperphosphatemia is a characteristic feature. Hyperphosphatemia can influence the occurrence of neurological complications such as stroke, dementia, and cognitive impairment. [34] As mentioned earlier, eccentricity is negatively correlated with functional integration, and small-worldness is associated with the efficiency of the brain network. Consequently, an increase in phosphate levels can be interpreted as an impairment of functional brain connectivity by reducing functional integration and compromising network efficiency.

This study has some limitations that warrant discussion. First, this study was conducted at a single center with an insufficient number of participants. Therefore, large-scale studies are required to obtain statistically significant results. Second, to reduce bias in the research results, fNIRS measurements at the same PD duration and additional information about patients, such as medication status, are required to properly interpret the research results. Since the comparison was made between patients with ESKD undergoing PD and healthy controls, the loss of functional brain connectivity may have been overestimated. To assess the impact of PD on functional brain connectivity in patients with ESKD, it would be helpful to measure and compare at the time point of PD initiation and subsequent pre-established time points. In addition, patients with any prior neurological or psychiatric diseases were excluded from the experimental group, but if imaging tests such as CT or MRI or vascular examinations had been conducted, underlying conditions that could introduce bias into the study could have been definitively excluded. Third, the NIRSIT Lite machine could only collect fNIRS data from the frontal lobe. However, because cognitive dysfunction is primarily associated with the frontal lobe, it is easy to interpret the results of this study. [35,36]

## 5. Conclusions

This study demonstrated significant alterations in functional brain connectivity in patients with ESKD undergoing PD. This suggests that functional brain connectivity was significantly reduced in patients with ESKD undergoing PD compared to healthy controls. Additionally, substantial correlations were observed between some clinical variables and measures of functional brain connectivity. The results of this study are expected to help explain the mechanism of neurological complications in patients with ESKD undergoing PD, allowing for a clearer understanding of the mechanism and discussion of treatment in future studies. In future studies, measuring changes in functional brain connectivity using fNIRS in ESKD patients undergoing HD and comparing them with patients with ESKD undergoing PD would help further understand the mechanisms involved.

## Supporting information

**S1 Table. Correlation analysis of clinical factors and functional brain connectivity.**
(DOCX)

## Acknowledgments

We thank the staff of nephrology department for their assistance in conducting this study.

## Author contributions

**Conceptualization:** Kang Min Park, Bong Soo Park.

**Data curation:** Dong Ah Lee, Yang Wook Kim, Junghae Ko.

**Formal analysis:** Yoo Jin Lee, Aryan Yohanes Djojo.

**Investigation:** Jiyae Yi, Sihyung Park.

**Methodology:** Jiyae Yi, Sihyung Park.

**Software:** Kang Min Park.

**Writing – original draft:** Chang Min Heo, Jiyae Yi.

**Writing – review & editing:** Chang Min Heo, Bong Soo Park.

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
