## [Decision Letter · Decision Letter 0]

1 Dec 2024

PONE-D-24-21021Analysis of functional brain connectivity in patient with end-stage kidney disease undergoing peritoneal dialysis using functional near infrared spectroscopyPLOS ONE

Dear Dr. Park,

Thank you for submitting your manuscript to PLOS ONE. After careful consideration, we feel that it has merit but does not fully meet PLOS ONE’s publication criteria as it currently stands. Therefore, we invite you to submit a revised version of the manuscript that addresses the points raised during the review process.

We look forward to receiving your revised manuscript.

Kind regards,

Ahmet Murt

Academic Editor

PLOS ONE

Journal Requirements:

 This research was supported by the 2023 Inje University research grant.  

Additional Editor Comments :

In this observational study, authors studied functional brain connectivity in patients with PD and compared with healthy controls. In addition to answering to the reviewers, please provide some more information about functional brain connectivity. Does it have a linear correlation with cognition? Are there any other ways to measure it? Is this one of the gold standard methods? Should it be supported by some clinical tests for cognitive function?

Reviewers' comments:

Reviewer's Responses to Questions

**Comments to the Author**

1. Is the manuscript technically sound, and do the data support the conclusions?

Reviewer #1: Yes

Reviewer #2: Yes

2. Has the statistical analysis been performed appropriately and rigorously? 

Reviewer #1: Yes

Reviewer #2: Yes

3. Have the authors made all data underlying the findings in their manuscript fully available?

Reviewer #1: Yes

Reviewer #2: Yes

4. Is the manuscript presented in an intelligible fashion and written in standard English?

Reviewer #1: Yes

Reviewer #2: Yes

5. Review Comments to the Author

Reviewer #1: The article is a study examining the neurological effects and brain connectivity in peritoneal dialysis patients, and it is one of the conditions that is frequently encountered in daily practice and is not always easy to manage. I have a few suggestions for this interesting study.

The patients were evaluated by a neurologist, Were any imaging modalities done? (CT? MRI?) I did not see any additional information in this regard.

Especially since prefrontal area data can be collected, have there been tests that can show dysfunction of this domain?

Were vascular examinations performed in both groups?

I was able to examine the biochemistry values of the patient group, but I could not see the healthy group. Was it not looked at or shared?

It is probably planned to increase the number, but I think it would strengthen the hypothesis if there was a third group receiving HD. I believe that these feedbacks should be made.

Reviewer #2: This is an interesting manuscript that gives us more information about the potential mechanisms of cognitive decay among people with end-stage kidney disease (ESKD) undergoing PD. The study is well conceived, but in my view, comparing people undergoing PD with healthy controls may overestimate the loss of brain connectivity. This could be addressed by evaluating people with ESKD at the onset of PD and then at pre-established time points, hence making each participant her or his control. As it is unlikely to be feasible to recruit and follow up participants for extended periods, it should be mentioned as a limitation and a window for follow-up studies.

6. PLOS authors have the option to publish the peer review history of their article (what does this mean? ). If published, this will include your full peer review and any attached files.

**Do you want your identity to be public for this peer review?** For information about this choice, including consent withdrawal, please see our Privacy Policy .

Reviewer #1: No

Reviewer #2: **Yes: ** Sergio I Valdés Ferrer

---

## [Author Response · Author response to Decision Letter 0]

4 Jan 2025

We would like to thank for the editor and reviewers of the PLOS One for taking their time to review our article. We made some corrections and clarifications in the manuscript after going over the reviewers’ comments, again.

We hope the revised manuscript will better meet the requirements of the PLOS One for publication. We would like to thank you once again for the constructive review by the reviewers and editor.

#Reviewer 1

The article is a study examining the neurological effects and brain connectivity in peritoneal dialysis patients, and it is one of the conditions that is frequently encountered in daily practice and is not always easy to manage. I have a few suggestions for this interesting study.

The patients were evaluated by a neurologist, Were any imaging modalities done? (CT? MRI?) I did not see any additional information in this regard. Especially since prefrontal area data can be collected, have there been tests that can show dysfunction of this domain? Were vascular examinations performed in both groups?

: Thank you for your wonderful review. Unfortunately, our study did not include imaging modalities such as CT or MRI. However, patients with any prior neurological or psychiatric diseases were excluded from the experimental group. If imaging tests had been conducted, we believe we could have more definitively ruled out neurological issues. This is discussed in the limitations.

The MoCA-K used in this study is a test that reflects dysfunction in the prefrontal area. While it is a screening tool for cognitive impairment, it includes several items that assess executive function and attention, which can reflect the function of the frontal lobe. We recruited patients without a neurological history, and the average MoCA-K score was 24, which is above the threshold of 22 that suggests cognitive impairment, and no correlation with functional brain connectivity was observed. Vascular examination was not conducted in either group, which is also mentioned as a limitation.

I was able to examine the biochemistry values of the patient group, but I could not see the healthy group. Was it not looked at or shared? It is probably planned to increase the number, but I think it would strengthen the hypothesis if there was a third group receiving HD. I believe that these feedbacks should be made.

: The control group consisted of individuals without underlying diseases and with no health issues. Since the study aimed to investigate the correlation between laboratory data and functional brain connectivity in patients with ESKD undergoing PD, laboratory data from healthy controls were not collected.

By measuring changes in functional brain connectivity using fNIRS in ESKD patients undergoing hemodialysis and comparing them, it would be possible to better understand the mechanisms through which PD affects neurological complications in ESKD patients in comparison to HD. I agree with your opinion, and this has been added as a limitation in the study.

#Reviewer 2

This is an interesting manuscript that gives us more information about the potential mechanisms of cognitive decay among people with end-stage kidney disease (ESKD) undergoing PD. The study is well conceived, but in my view, comparing people undergoing PD with healthy controls may overestimate the loss of brain connectivity. This could be addressed by evaluating people with ESKD at the onset of PD and then at pre-established time points, hence making each participant her or his control. As it is unlikely to be feasible to recruit and follow up participants for extended periods, it should be mentioned as a limitation and a window for follow-up studies.

: Thank you for your excellent review. I have added the additional points to the limitations section based on your advice.

---

## [Editor Report · Decision Letter 1]

6 Apr 2025

Analysis of functional brain connectivity in patient with end-stage kidney disease undergoing peritoneal dialysis using functional near infrared spectroscopy

PONE-D-24-21021R1

Dear Dr. Park,

We’re pleased to inform you that your manuscript has been judged scientifically suitable for publication and will be formally accepted for publication once it meets all outstanding technical requirements.

Kind regards,

Keiko Hosohata, Ph.D.

Academic Editor

PLOS ONE
---

## [Editor Report · Acceptance letter]

PONE-D-24-21021R1

PLOS ONE

Dear Dr. Park,

I'm pleased to inform you that your manuscript has been deemed suitable for publication in PLOS ONE. Congratulations! Your manuscript is now being handed over to our production team.

Kind regards,

on behalf of

Dr Keiko Hosohata

Academic Editor

PLOS ONE